# Examining English- and Spanish-Speaking Therapist Behaviors in Parent–Child Interaction Therapy

**DOI:** 10.3390/ijerph19084474

**Published:** 2022-04-08

**Authors:** Yessica Green Rosas, Kristen M. McCabe, Argero Zerr, May Yeh, Kristine Gese, Miya L. Barnett

**Affiliations:** 1Department of Counseling, Clinical, & School Psychology, University of California, Santa Barbara, CA 93106, USA; mbarnett@ucsb.edu; 2Department of Psychological Sciences, University of San Diego, San Diego, CA 92110, USA; kmccabe@sandiego.edu (K.M.M.); kgese@sandiego.edu (K.G.); 3Child and Adolescent Services Research Center, San Diego, CA 92123, USA; myeh@sdsu.edu; 4Psychology Program, California State University, Channel Islands, Camarillo, CA 93012, USA; argero.zerr@csuci.edu; 5Department of Psychology, San Diego State University, San Diego, CA 92120, USA; 6Department of Psychiatry, University of California, San Diego, CA 92093, USA

**Keywords:** parent–child interaction therapy, behavioral parent training, language, therapist coaching behaviors

## Abstract

Parent–child interaction therapy (PCIT) is a best-practice treatment for behavior problems in young children. In PCIT, therapists coach parents during in-vivo interactions to strengthen the parent–child relationship and teach parents effective ways of managing difficult child behaviors. Past research has found that different therapist coaching styles may be associated with faster skill acquisition and improved parent engagement. However, most research examining therapist behaviors has been conducted with English-speaking families, and there is limited research examining therapist behaviors when working with Spanish-speaking clients. In this study, English- and Spanish-speaking therapists’ coaching behaviors (e.g., directive versus responsive) were examined, as well as their association with client outcomes, including speed of parental skill acquisition and treatment completion. Results suggested that coaching styles varied significantly between sessions conducted in Spanish versus English. In Spanish sessions, therapists had more total verbalizations than in English sessions and demonstrated higher rates of both total directive and responsive coaching. Responsive coaching was found to predict treatment completion across groups, while directive coaching was not. Directive and responsive coaching were not found to predict the rate of parental skill acquisition. Implications regarding the training of therapists and emphasizing cultural considerations are discussed.

## 1. Introduction

Youth behavior problems are highly prevalent, negatively impact youth and families, and have been found to be persistent if left untreated [1,2]. Long-term outcomes of early-onset conduct problems include later psychopathology, deviant behavior, poor academic success, and high societal costs (e.g., involvement in the justice system [3,4,5,6]). Unfortunately, low-income and ethnic minority families are at an increased risk for mental health issues given that they often face heightened levels of stressors, which may be due to financial burdens, immigration status, ethnic and racial discrimination, and prejudice, as well as systemic and structural barriers to access to and engagement in proper mental health treatments [7,8,9]. Furthermore, most Latinx youth in the United States report experiencing at least one childhood traumatic event, with 30% experiencing four or more traumatic events in their life, putting them at high risk for mental health issues [10]. Despite the high need for services, disparities in access to, and engagement in, proper mental health care have also been reported for black and Latinx youth [7,11,12,13]. It is important for efforts to be focused on reducing these disparities, in order to mitigate negative outcomes for ethnic minority children and families. 

Extensive literature has supported the efficacy of behavioral parent training interventions (BPTs) in addressing childhood behavioral problems [14]. Treatment providers delivering BPTs teach parents a wide range of techniques, including positive reinforcement, (e.g., praising their child), as well as natural consequences (e.g., non-violent disciplining techniques), which ultimately help to reduce children’s disruptive behaviors [15]. Parent–child interaction therapy (PCIT) is an evidence-based BPT that has been shown to significantly reduce externalizing behavioral problems in young children, ages 2–7 [16]. It is currently considered a best-practice treatment approach for externalizing behavioral problems [14], as well as a promising potential transdiagnostic behavioral intervention, given that it has been shown to be efficacious in treating behavioral problems for children with anxiety, depression, ADHD, and autism [17,18,19,20]. Families who have participated in PCIT have been found to keep treatment gains up to 3–6 years after treatment completion [21].

PCIT is unique in that it uses in-vivo coaching as a major component of treatment, which allows parents to get real-time feedback from their therapist while they interact with their child. A meta-analysis of BPTs found that programs with in-vivo coaching have larger treatment effect sizes in comparison to those without this component [22]. A typical PCIT session involves therapists coaching parents via a Bluetooth microphone as they implement parenting skills that are taught in treatment [16]. Through coaching, parents practice and strengthen the behavioral skills taught by their therapist, while they interact and strengthen their relationship with their child during different play situations. Treatment is split up into two phases. The first phase is called the child-directed interaction (CDI) phase and consists of teaching parents positive parenting skills, which they practice during in-session coaching, as well as at home. These skills include labeled praises, behavioral descriptions, and reflections of child verbalizations. The second phase of treatment is called the parent-directed interaction (PDI) phase, in which parents learn about and deliver effective discipline strategies. During both phases of treatment, parents are coached by therapists as they practice implementing parenting skills with their children. 

Given that coaching is a core component of PCIT and has been associated with larger effect sizes, research has begun to investigate the types of coaching statements that lead to improved outcomes for parents and children [23]. The therapist–parent interaction coding system (TPICS) was developed to investigate different coaching styles related to improved parental skill development in PCIT [23,24]. The TPICS measures two categories of coaching styles—directive coaching and responsive coaching. Directive coaching techniques happen before parent verbalizations and tell a parent to use specific skills. Examples of these include commands (e.g., praise her for sharing), prompting (e.g., “thank you for….”), and modeling of skills. Responsive coaching techniques happen after a parent’s verbalization to reinforce what the parent said. Examples of responsive coaching include praises of the skills used (e.g., “great labeled praise”), reflective descriptions (e.g., “that was an unlabeled praise”), and process comments, which connect a child’s behaviors with the parents (e.g., “those behavior descriptions are helping him focus”). The TPICS was also developed to identify the parenting skills (e.g., behavior descriptions) that a therapist targeted in their coaching. This allows for the measure to identify how therapist coaching verbalizations relate to the skills that parents use during session. 

Past research using the TPICS has investigated how coach behaviors in early CDI sessions related to parent skill acquisition and engagement in treatment [23]. In the first study validating the TPICS, responsive coaching was found to mediate parental skill acquisition between one session and the next, while the use of directive coaching was not [24]. Another study, conducted with English- and Spanish-speaking families, similarly found that skill acquisition between two sessions of PCIT was impacted by responsive coaching [25]. Barnett et al. (2017) found that therapist coaching not only impacted parent skill acquisition between sessions, but that it was also associated with treatment dropout. Specifically, therapists seeing families who dropped out of treatment were more likely to use fewer responsive techniques and more drills, which is a directive technique. This study was conducted with a predominantly English-speaking Latinx population. 

While most of the evidence base for PCIT is with non-Hispanic, white families, past research has demonstrated that families from diverse backgrounds may also benefit from this treatment model [26,27,28]. Efforts have also been successful in adapting PCIT for ethnic minority families. For example, a culturally modified version of PCIT, called Guiando a Niños Activos (GANA), found promising results in treating behavioral problems for Mexican-American children [29]. A personalized version of PCIT (MY PCIT) has also been developed, in which specific Parent Explanatory Model parameters that have been found to impact treatment engagement and outcomes can be assessed and addressed in treatment for African-American, Asian-American, Hispanic, and non-Hispanic white families [30]. 

PCIT was developed in English, and most of the evidence base available examining its efficacy is from studies conducted with English-speaking families. While PCIT has been found to be effective in treating behavioral problems with ethnic minority youth, including Mexican-American families, few studies have examined how treatment, and specifically coaching, may look different when working with Spanish speakers [25]. In the one study that has examined coaching behaviors in Spanish, Heymann et al. (2020) found that Spanish-speaking therapists used significantly fewer responsive coaching in comparison to English-speaking therapists. 

It is crucial that coaching in Spanish be further examined, given that Latinx families may experience higher levels of barriers to engagement, including a lack of therapists who can provide quality services in their native language. Furthermore, as of 2019, over 41 million Spanish speakers were estimated to reside in the US [31]. Because there is a dearth of literature examining Spanish treatment delivery, we are left with less available resources to inform proper training of Spanish-speaking therapists to deliver effective treatment to Spanish-speaking clients. Given that coaching is a primary component of PCIT and coaching styles have been found to be associated with skill acquisition, it is of particular interest to examine whether there may be differences in coaching between English- and Spanish-speaking therapists. Additionally, it is important to understand how coaching may relate to client outcomes for English- and Spanish-speaking clients. These findings can help inform strategies to enhance treatment outcomes for Spanish-speaking families, in order to decrease mental health service disparities.

The current study examined coaching from 49 pre-recorded PCIT sessions from two previous studies examining adapted versions of PCIT for culturally diverse families. We sought to address the following study objectives: (1) identify differences in responsive and directive coaching for English- and Spanish-speaking therapists; (2) investigate how coaching styles predict the rate of parental skill acquisition; and (3) examine how coaching styles relate to client dropout.

## 2. Materials and Methods

### 2.1. Data Set

The data set consisted of video recordings and parent-report measures of participants from two PCIT studies. The first study separated participants into groups of three different conditions: (1) a culturally adapted version of PCIT (GANA), (2) PCIT, and (3) treatment-as-usual [32]. We examined data from participants assigned to either the GANA or PCIT conditions (*n* = 40), and all participants in this study were Mexican-American. The second study from which we examined participant sessions included 32 families from four different ethnic groups: African-American, Asian-American, Latinx, and non-Hispanic white [30]. In both of these studies, the core components of treatment were maintained, including therapist coaching. Participants for whom there was a viable video recording of the second CDI coach session were included in our study. In total, our data included one session from 49 participants across the two studies, due to missing or corrupted videos for the remaining participants. In the sample, there were 30 English-speaking sessions, 16 Spanish-speaking sessions, and 3 bilingual English-Spanish sessions. Bilingual session videos were grouped with English sessions in the analyses given the low frequency of Spanish verbalizations in the sessions.

### 2.2. Eligibility for GANA Study

The research team advertised the study by conducting outreach to programs, including Head Start, community mental health, and family resource centers, to encourage referrals. The study received referrals from various sources: 42% school or teachers; 22% other agency, clinic, or hospital; 12% self-referred; 5% friend; 2% physician; and 2% social worker. Families were screened for eligibility for the study by phone and determined to be eligible if they identified their child as: (1) Mexican-American; (2) between the ages of 3 and 7; (3) without a diagnosis of autism, intellectual disability, or psychosis; (4) not participating simultaneously in any other psychosocial treatment for the child’s behavior problems; and (5) with clinically significant behavior problems as measured by parent report on the Eyberg Child Behavior Inventory Intensity Scale (ECBI [33]). Screeners for participants were available in both English and Spanish.

### 2.3. Eligibility for MY PCIT Study

The research team advertised the study in the community via word of mouth, flyers, social media, and recruitment by other community agencies that serve children and parents. Families could enter the study via two possible avenues: (1) by seeking services at one of the participating community service agencies providing outpatient psychotherapy, through which an initial screening by agency staff would be conducted for appropriateness for the study/PCIT and permission for study contact would be gathered, or (2) by contacting the study directly. Families were eligible for the MY PCIT study if (1) the child was between the ages of 2–7; (2) child was above the clinical cut-off (raw score of 131 or above) on the ECBI, (3) child was identified by the parent as Asian-American, black/African-American, Latinx, and/or non-Hispanic white, and (4) parents were English or Spanish speaking. Families were excluded from the study if they were simultaneously receiving other therapy for behavioral problems or the child or primary caregiver had an intellectual disability or autism. 

### 2.4. Participants

Participants were families seeking treatment at a community mental health clinic for a young child (age 2–7) with clinically significant behavior problems. In the current sample (*N =* 49), sixty-seven percent of children were male and 33% were female. On average, children were 4.39 years old. Eighty percent of families (*n* = 39) identified their child as belonging to one racial/ethnic group: 8% as Asian-American (*n* = 4), 6% as African-American (*n* = 3), 51% as Latinx (*n* = 25), and 14% as non-Hispanic white (*n* = 7). Twenty percent of families (*n* = 10) identified their child as belonging to multiple racial/ethnic groups. All families from the GANA sample were Mexican-American, given that this was a criterion for study inclusion.

All caregivers in the sample were mothers (*M* = 35.55 years). Three fathers and five other caregivers also participated in treatment, though their sessions were not coded. Caregivers included 6 Asian-American, 2 African-American, 28 Latinx, 9 non-Hispanic white, and 4 families belonging to multiple racial/ethnic groups. In regard to the highest education obtained, 18% of caregivers did not graduate high school, 10% graduated from high school, 33% had some college or technical school, 22% graduated from college, and 16% obtained a graduate school degree.

### 2.5. GANA and MY PCIT Study Procedures 

Eligible families for the GANA and MY PCIT studies were invited to complete a 3–4 h in-person assessment, where they completed informed consent forms and a clinical interview. During this assessment, parents completed measures about themselves (e.g., parenting stress) and their child, as well as a behavioral observation of the parent–child interaction (see Measures, below). Families from both studies completed a post-treatment assessment, as soon as possible, after treatment termination. Parents were paid 100 USD for each of the assessments.

### 2.6. Current Study Procedure 

A team of four research assistants were trained on the empirically validated, Therapist–Parent Interaction Coding System (TPICS [24]), and Dyadic Parent–Child Interaction Coding Systems (DPICS [34]). Videotapes of 5-min segments of the second CDI coaching session, in line with what was conducted in the original TPICS study [24], from 49 participants of the GANA and MY PCIT studies, were digitized and transcribed. Therapist and parent verbalizations in treatment video recordings were coded by research assistants, and a sample of 25% of tapes were double-coded, in order to examine interrater reliability. Meetings were conducted with all researchers and supervisors on a weekly basis, in order to discuss questions that came up for coders and to avoid coder drift. 

## 3. Measures

### 3.1. Therapist–Parent Interaction Coding System

The TPICS [24] is a behavioral observation coding system that assesses therapist coaching statements. Each therapist verbalization is coded in the TPICS to identify the technique used (e.g., modeling), as well as the targeted parent skill (e.g., reflection). Coaching technique codes are categorized as being directive (e.g., direct command, modeling, prompting, drills), responsive (e.g., labeled praise, reflective description, process comment), neutral (i.e., talk), or critical (i.e., corrective criticism). Targeted parenting behaviors that therapists coach include labeled praises, reflections, behavior descriptions, questions, commands, negative talks, and other (i.e., additional behaviors targeted in PCIT but not coded by the DPICS-IV, including ignoring, imitation, and enjoyment). The TPICS also includes the code, mistake, if the therapist incorrectly coaches a targeted parenting behavior (e.g., praising a parent for using a behavior description when they actually said a reflection). Codes used in the TPICS are included in Table 1. 

In past research, the TPICS has been found to have excellent reliability and can predict parents’ skill acquisition from one session to the next [24], as well as parent completion of treatment [23]. In the current study, the TPICS was used to code therapist verbalizations, in order to examine and compare rates of coaching techniques used by English- and Spanish-speaking therapists, as well as the parenting skills that were targeted by therapists in session. All therapist verbalizations were coded, and frequencies were summed of directive coaching statements, responsive coaching statements, neutral verbalizations, and total coaching statements (i.e., a composite of all types of coaching). Interrater reliability for the therapist technique codes was 82% and was 74% for targeted parenting behaviors. 

### 3.2. Dyadic Parent–Child Interaction Coding System

The Dyadic Parent–Child Interaction Coding System-IV [34] is a behavioral observation coding system that was designed to assess the quality of parent–child interactions. The DPICS was used to code parent skills used during the coaching portion of the coaching session to help identify if Spanish- and English-speaking parents used different frequencies of verbalizations while they were being coached. Codes used in this study included the parenting skills that are the focus of treatment (i.e., “Do” skills), including labeled praises, behavioral descriptions, and reflections, as well as verbalizations taught to be decreased during treatment (i.e., “Don’t” skills), including questions, commands, and negative talks. Additionally, neutral talk was coded (e.g., the parent describing their own behavior). Interrater reliability for parent verbalizations was 89%. 

### 3.3. Eyberg Child Behavior Inventory 

The ECBI [33] is a 36-item parent-rating scale of conduct problems for children between the ages of 2 to 16. Parents rate the frequency of each disruptive behavior on a 7-point Likert scale from 1 (never) to 7 (always), which are summed to yield the Intensity Scale. The ECBI is sensitive to treatment effects and has excellent internal reliability (α = 0.92–0.95 [35]). The ECBI was used to determine participant eligibility in the two studies from which the current sample is composed and was included as a covariate in analyses related to how coaching impacts parent skill acquisition and completion of treatment.

### 3.4. Parenting Stress Index-SF

The parenting stress index is a 36-item questionnaire designed to evaluate stress in the parent–child system (PSI [36]). The PSI demonstrated acceptable internal consistency (a = 0.74–0.88) and test-retest reliability [37]. Pre-treatment PSI scores were used as a covariate to control for levels of parental stress.

### 3.5. Parental Skill Acquisition

Rate of parental skill acquisition was measured by the number of sessions required for families to reach graduation criteria in the CDI phase of treatment. In order to reach criteria in the CDI phase of treatment, parents must be able to use 10 labeled praises, 10 behavioral descriptions, and 10 reflections of child verbalization, and less than 3 questions, commands, or criticisms during a 5 min play interaction with their child. Families who dropped out prior to reaching CDI graduation criteria (*n* = 5) were excluded from the current study’s analyses examining parental skill acquisition.

### 3.6. Treatment Completion

Treatment completion was measured using the graduation criteria defined by PCIT standards. This includes reaching parent skill criteria for the CDI and PDI phases of treatment, having an ECBI score at or below 114, and ensuring parental comfort and confidence in handling child behaviors.

### 3.7. Therapist and Parent Verbalizations Statistical Analyses 

Overall rates of directive and responsive coaching, as well as neutral verbalizations used in coaching that did not fall into either directive or responsive categories, were compared between English and Spanish coaching sessions using independent samples *t*-tests. Differences between English- and Spanish-speaking clients were also examined using independent samples t-tests, including parent Do-skills, Don’t skills, and overall verbalizations. 

### 3.8. Client Outcomes Statistical Analyses 

Two regression models were run, in order to examine the relationship between directive and responsive therapist coaching and client outcomes. Model A (a linear regression model) examined directive coaching and responsive coaching as predictors for the rate of parental skill acquisition, as measured by the number of CDI sessions; Model B (a logistic regression model) examined directive coaching and responsive coaching as predictors of treatment completion versus treatment dropout. Covariates for each model included language (Spanish and English), ECBI intensity score at pre-treatment, and PSI score at pre-treatment. Condition was not included as a covariate, given that it broke the threshold for multicollinearity with the Language covariate.

## 4. Results

### 4.1. Verification of Assumptions

Before running analyses, the assumption of homogeneity of variance was analyzed by a Levene’s test of independence for all coaching techniques, targeted verbalizations, and parent verbalizations; the appropriate p-values (equal variance not assumed) were used for the variables that were indicated to have violated the homogeneity of variance assumption. All assumptions for the regression models were within an acceptable range and did not require any transformations.

### 4.2. Differences in Coaching Techniques 

Independent sample t-tests comparing coaching techniques indicated that coaching styles varied significantly between sessions conducted in Spanish and English for all composite coaching categories. Composite scores included total responsive verbalizations, total directive verbalizations, and total verbalizations (i.e., directive verbalizations, responsive verbalizations, neutral verbalizations, and corrective criticisms). Spanish-speaking therapists had significantly more total verbalizations than English-speaking therapists (Spanish: *M* = 69.25, *SD* = 15.33; English: *M* = 42.70, *SD* = 13.47), *t*(21) = 2.47 *p* < 0.001. In Spanish sessions, therapists were more likely to use higher rates of directive coaching statements overall (*M* = 28.81, *SD* = 15.11) than in English sessions (*M* = 16.36, *SD* = 8.57), *p* = 0.006, as well as higher rates of responsive coaching (Spanish: *M* = 29.19, *SD* = 12.76; English *M* = 20.82, *SD* = 8.13), *t*(21) = 2.40, *p* = 0.026. 

Regarding specific techniques, in Spanish sessions, therapists used significantly more unlabeled praises (Spanish: *M* = 11.06, *SD* = 5.22, English: *M* = 3.67, *SD* = 3.29), *t*(21) = 5.19, *p* < 0.001, and neutral talks (Spanish: *M* = 10.38, *SD* = 6.70, English: *M* = 4.21, *SD* = 4.80), *t*(47) = 3.69, *p* = 0.001. In English sessions, therapists used significantly more prompting (Spanish: *M* = 0.00, *SD* = 0.00, English: *M* = 0.27, *SD* = 0.45), *t*(32) = 3.46, *p* =.002.

In terms of the verbalizations targeted by therapists during treatment, Spanish-speaking therapists were found to target questions (Spanish: *M* = 2.25, *SD* = 3.04, English: *M* = 0.27, *SD* = 0.67), *t*(47) = 0.26, *p* = 0.02, unlabeled praises (Spanish: *M* = 3.81, *SD* = 4.39, English: *M* = 1.00, *SD* = 1.46), *t*(47) = -2.50, *p* = 0.02, and “other” verbalizations (i.e., verbalizations that are not do or don’t skills) (Spanish: *M* = 15.88, *SD* = 8.52, English: *M* = 8.52, *SD* = 7.80), *t*(47) = 3.01, *p* = 0.004) significantly more than English-speaking therapists. See Table 1 for comparisons of English and Spanish coaching techniques and categories. 

### 4.3. Differences in Parent Behaviors 

Results from parent verbalizations found that Spanish-speaking parents also had significantly higher rates of verbalizations during sessions (Spanish: *M* = 68.69, *SD* = 23.09; English: *M* = 53.94, *SD* = 17.70), *p* = 0.017. Specifically, Spanish-speaking parents gave significantly more direct commands to their children (Spanish: *M* = 4.50, *SD* = 4.91) than English-speaking parents (*M* = 1.30, *SD* = 1.99), *p* = 0.022. They also gave more unlabeled praises (Spanish: *M* = 9.75, *SD* = 6.27; English: *M* = 4.73, *SD* = 3.67), *p* = 0.001. Other parenting skills, as coded by the DPICS, were not significantly different (see Table 1). 

### 4.4. Predicting Client Outcomes

Model A, examining directive and responsive coaching as predictors of the rate of parent skill acquisition as measured by number of CDI sessions, was not significant (*F* [5, 43] = 0.95, *p* = 0.46; *R*^2^ = 0.11). Model B, examining directive and responsive coaching as predictors of treatment completion versus treatment drop out was found to be significant, (χ^2^ [5] = 16.83, Nagelkerke *R*^2^ = 0.40, *p* = 0.005; 73.5% accurate classification) in that responsive coaching positively predicted treatment completion (*B* = 0.12, Wald = 5.63, *p* = 0.02, *OR* = 1.13), while directive coaching did not predict completion (*B* = −0.04, Wald = 1.15, *p* = 0.28, *OR* = 0.96). For every additional responsive verbalization, there was a 14.6% increased chance of completing treatment. Independent samples t-tests, which did not include covariates, found that there was not a statistically significant difference in rates of directive verbalizations between treatment completers (*M* = 18.75) and non-completers (*M* = 23.59; *t*[47] = 1.30, *p* = 0.20); however, treatment completers had significantly higher rates of responsive verbalization (*M* = 26.75), compared to treatment non-completers (*M* = 17.53; *t*[47] = 3.19, *p* = 0.003).

## 5. Discussion

Although past research has demonstrated the critical role that therapist coaching can have on client engagement and outcomes in PCIT [23,24,25], there is currently a dearth of literature in this topic area, especially for Spanish-speaking families. This study contributes to the literature by (1) examining and comparing both English- and Spanish-speaking therapists’ use of directive and responsive coaching, and (2) expanding on previous studies by examining the relationship between different coaching styles and rate of parental skill acquisition and treatment completion.

### 5.1. English- and Spanish-Speaking Therapist Coaching

This study found that overall Spanish-speaking therapists had more total verbalizations, including higher rates of both responsive and directive coaching, along with statements that did not fall into either of these categories. Spanish-speaking therapists were found to use higher rates of unlabeled praises. The use of labeled praises, as opposed to unlabeled praises, may be a point of intervention for training of Spanish-speaking clinicians. Unlabeled praises show warmth, support, and enthusiasm. However, using more specificity, as is the case with clear labeled praises, may better facilitate skill acquisition for parents. To date, there has only been one other study examining coaching differences between English- and Spanish-speaking therapists [25]. Results from this previous study also found that English- and Spanish-speaking therapists significantly differed in their use of both directive and responsive coaching. However, English-speaking therapists in the previous study sample were found to use higher rates of responsive coaching, while, in the current sample, the opposite was true. It should be noted that samples from these two studies were taken from two distinct regions in the United States, and that the current study examined coaching conducted in PCIT, while the previous study examined coaching from the brief parenting intervention based on PCIT, the infant behavior program (IBP). Additionally, in the current study, coaching was provided by community clinicians, whereas graduate student clinicians provided treatment in the Heymann et al., (2020) study. The contradictory findings suggest that various factors beyond language spoken may impact therapist verbalizations. Further research is needed, in order to examine how these initial findings may relate to therapists practicing in various regions, the demographics of the clients they serve, and the training and supervision they receive. It is also important to consider that therapists may be responding to different parent characteristics, and so it is important to further examine the transactional interaction between therapists and parents during session.

### 5.2. Parent Verbalizations

Parent use of skill during a session is important to understand as it may impact the coaching statement that therapists use. That is to say, for therapists to use responsive coaching statements, the parent has to first use the skill that is being reinforced. Therefore, in this study, parent verbalizations were coded to understand if differences between Spanish and English therapists were being impacted by differences in the parents’ behaviors. Significant differences were found between English- and Spanish-speaking parents. Spanish-speaking parents were found to use higher rates of direct commands. This is in line with previous research finding that language predicted parents’ level of “Don’t” skills, with Spanish-speaking parents more frequently using direct commands [38]. Spanish-speaking parents have been found to use higher levels of direct commands in comparison to their English-speaking counterparts, and past literature has offered the possibility of this being due to parents being higher on the control dimension of parenting [38,39]. This may be helpful in the second phase of treatment, where caregivers are taught to use direct commands with their children. However, Spanish-speaking parents may require additional support during the first phase of treatment, given that directive statements, including direct commands, must be avoided during this phase. We also found that Spanish-speaking parents used significantly more unlabeled praises than English-speaking parents while they were being coached. However, there were no significant differences in the amount of labeled praises used by English versus Spanish-speaking parents, which are traditionally encouraged over unlabeled praises during treatment. We also found that in sessions conducted in Spanish, therapists targeted unlabeled praises more often than in sessions conducted in English. Examples of this included Spanish-speaking therapists modeling unlabeled praises (e.g., *you are so smart*). This points to a potential area to focus on when training Spanish-speaking therapists, which is reducing the use of unlabeled praises in coaching, so that parents use them less frequently. Future directions for this project will include a closer examination of the two-way interaction between parent use of skills and therapist coaching. 

### 5.3. Client Outcomes

The current study found that directive and responsive coaching did not predict the rate of parental skill acquisition as measured by the number of CDI sessions. Previous studies have found that responsive coaching predicted higher levels of positive parenting skills from one session to the next and the length of the CDI phase of treatment [23,24,25]. Though the speed of skill acquisition appears to relate to responsive coaching in previous research, our study suggests that additional research is needed to identify additional factors that might account for this change. 

The current study did find that responsive coaching positively predicted parent completion of treatment, while directive coaching did not. This is in line with previous research, finding that less responsive coaching and higher rates of a directive skill (i.e., drills) were used with families who had dropped out of treatment [23]. The current study expands on these findings by examining outcomes with both English- and Spanish-speaking families, including representation from a diverse sample of Asian-American, African-American, Latinx, and non-Hispanic white families. However, although the current study includes families from various backgrounds, most Spanish-speaking families in the sample were solely of Mexican-American descent, which serves as a potential limitation. More research should be conducted to better understand how coaching styles may impact client outcomes for English-speaking families from different regions, as well as Spanish-speaking families from other ethnic groups. 

### 5.4. Strengths and Limitations

There are various strengths that should be highlighted in this current project. Our findings expand an under-examined area of study: therapist coaching. This is important given that coaching is a primary component of treatment in PCIT and has been found to be associated with larger effect sizes in treatment. Our study also included a diverse sample with representation from African-American, Asian-American, Latinx, and non-Hispanic white families. Furthermore, Spanish coaching in PCIT had only been examined in one other study [25]. This project expands on previous research findings, further contributing to the literature.

This study also includes various limitations to be considered. Given that we used archival data when examining treatment sessions in coaching, there were a number of video recordings that could not be analyzed due to technological or other logistical issues. Increased sample sizes with all videos included would be valuable for future research on therapist coaching. We also did not examine the two-way interaction between therapist and parent, which leaves much information regarding the effects that each party’s verbalizations have on each other to be explored. Finally, all Spanish-speaking families in our current study identified as Mexican-American. Thus, results found from this sample may not generalize to Spanish-speaking families from other cultural backgrounds. More research is needed, in order to further examine linguistic differences found in treatment, as well as how these relate to outcomes for families from different racial and ethnic backgrounds.

## 6. Conclusions

Continuous efforts have been made to decrease existing mental health disparities for ethnic minority families. By examining differences in current English and Spanish provision of services, we can better understand factors that may be contributing to positive client outcomes. This, in turn, can inform culturally sensitive training of bilingual and bicultural treatment providers in order to enhance access to culturally responsive treatment and help to mitigate racial and ethnic mental health disparities. The findings in this study highlight the importance of following parent’s lead during treatment and focusing on reinforcing strategies when teaching parents new skills, rather than emphasizing directive approaches in treatment. There have been very few studies that examine session-level differences between English and Spanish sessions, and future research should examine specific efforts that Spanish-speaking therapists make, in order to retain families in BPTs, despite the various systemic and structural barriers that many ethnic minority families face. This strengths-based approach would allow for continued research examining the specific factors that facilitate high quality services for culturally diverse clients.

## Figures and Tables

**Table 1 ijerph-19-04474-t001:** Therapist and Parent Verbalizations.

	English		Spanish			
	*M*	*SD*	*M*	*SD*	*t*	*p*-Value
** Directive Therapist Codes **						
Modeling	7.73	5.48	13.63	11.47	1.95	0.066
Prompting	0.27	0.452	0.00	0.00	3.46	0.002 **
Direct Command	2.21	4.04	5.75	7.34	1.80	0.087
Indirect Command	3.85	3.33	5.25	4.19	1.27	0.211
Drill	0.06	0.35	0.00	0.00	0.69	0.492
Child Observation	2.24	2.00	4.19	4.93	1.52	0.147
**Total Directive**	16.36	8.57	28.81	15.11	3.06	0.006 **
** Responsive Therapist Codes **						
Labeled Praise	14.52	7.03	13.06	8.90	0.62	0.537
Unlabeled Praise	3.67	3.29	11.06	5.22	5.19	0.000 ***
Reflective Description	1.33	1.71	2.19	1.56	1.69	0.098
Process Comment	0.85	1.30	1.31	2.02	0.84	0.412
**Total Responsive**	20.82	8.13	29.19	12.76	2.40	0.026 *
** Other Therapist Codes **						
Talk	4.21	4.80	10.38	6.70	3.69	0.001 **
Corrective Criticism	0.45	0.87	1.56	2.37	1.82	0.087
** Total Therapist Codes **	42.70	13.47	69.25	15.33	6.19	0.000 ***
** Targeted Parenting Behavior **						
Talk	0.85	3.06	0.69	1.35	0.20	0.842
Reflection	4.33	3.42	4.81	4.75	0.40	0.688
Question	0.27	0.674	2.25	3.04	2.57	0.021 *
Behavior Description	7.45	4.66	8.69	5.40	0.83	0.414
Direct Command	0.03	0.174	0.06	0.250	0.53	0.602
Indirect Command	0.09	0.29	0.19	0.40	0.96	0.344
Negative Talk	0.00	0.00	0.00	0.00	--	--
Unlabeled Praise	1.00	1.46	3.81	4.39	2.50	0.023 *
Labeled Praise	6.79	4.69	4.44	3.46	1.78	0.082
Mistake	2.15	2.28	2.69	3.03	0.69	0.492
Other	8.52	7.80	15.88	8.52	3.01	0.004 **
**Total TPB Codes**	25.36	10.19	34.00	19.00	1.70	0.105
** Parent Verbalizations (DPICS) **						
Talk	21.73	12.15	22.94	12.50	0.32	0.747
Reflection	4.67	3.35	4.19	3.78	0.45	0.655
Question	6.55	5.66	7.13	4.77	0.35	0.726
Behavior Description	7.18	4.42	12.38	10.11	1.97	0.065
Direct Command	1.30	1.99	4.50	4.91	2.51	0.022 *
Indirect Command	1.36	1.58	2.50	3.45	1.26	0.255
Negative Talk	0.70	1.10	0.44	1.03	0.79	0.435
Unlabeled Praise	4.73	3.67	9.75	6.27	3.54	0.001 **
Labeled Praise	5.73	3.80	4.88	2.58	0.81	0.423
**Total Parent Verbalizations**	53.94	17.7	68.69	23.09	2.47	0.017 *

* *p* < 0.05, ** *p* < 0.01, *** *p* < 0.001.

## Data Availability

Some data used in the preparation of this manuscript are available from the National Institute of Mental Health (NIMH) Data Archive (NDA). NIMH NDA is a collaborative informatics system created by the NIMH to provide a national resource to support the sharing of federally-funded data for accelerating research. Dataset identifier(s): 10.15154/1524699.

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
