# Peer review of "Examining English- and Spanish-Speaking Therapist Behaviors in Parent–Child Interaction Therapy"

_ijerph, 2022, doi:10.3390/ijerph19084474_

Round 1

Reviewer 1 Report

This is a well-written manuscript with wide interest to researchers. I have a few suggestions which I proceed to list:

  1. In the title it would be appropriate to break down the acronym PCIT.
  2. In the page 3, please express the questions also as objectives.
  3. The statistical techniques used (independent samples t test and regression) have certain assumptions that must be met (normality, linearity, independence of the residuals...). The fulfillment of these assumptions has not been analyzed. I recommend the authors to report the verification of assumptions and, if they are not met, use other more appropriate techniques (for example, non-parametric statistics). Please consider that your results may not be valid due to non-fulfilment of assumptions, another interesting option could be the use of bootstrapping.
  4. Review compliance with APA 7ed standards regarding to citation of 3 or more authors.

Author Response

We are very appreciative of your careful review of the manuscript, which is now titled, “Examining English- and Spanish-speaking Therapist Behaviors in Parent-Child Interaction Therapy.” The suggestions the reviewers made greatly strengthened the manuscript. We have made the changes in the document you sent with track-changes to ease in review. Responses are outlined below.

  1. In the title it would be appropriate to break down the acronym PCIT.

We have changed the title to, “Examining English- and Spanish-speaking Therapist Behaviors in Parent-Child Interaction Therapy.”

  1. In the page 3, please express the questions also as objectives.

We have changed the study questions to objectives (page 3, line 135). It states:

We sought to address the following study objectives: 1) Identify differences in responsive and directive coaching for English- and Spanish-speaking therapists.? 2) Investigate how coaching styles predict the rate of parental skill acquisition. And 3) Examine how coaching styles relate to client dropout.

  1. The statistical techniques used (independent samples t test and regression) have certain assumptions that must be met (normality, linearity, independence of the residuals...). The fulfillment of these assumptions has not been analyzed. I recommend the authors to report the verification of assumptions and, if they are not met, use other more appropriate techniques (for example, non-parametric statistics). Please consider that your results may not be valid due to non-fulfilment of assumptions, another interesting option could be the use of bootstrapping.

We have checked assumptions and added the following section (Page 8, Lines 296-302)

4.1. Verification of Assumptions

       Before running analyses, the assumption of homogeneity of variance was analyzed by a Levene’s test of independence for all coaching techniques, targeted verbalizations, and parent verbalizations, and the appropriate p-values (equal variance not assumed) were used for the variables that were indicated to have violated the homogeneity of variance assumption. All assumptions for the logistic regression models were within an acceptable range and did not require any transformations.

  1. Review compliance with APA 7ed standards regarding to citation of 3 or more authors

We have changed all citations and reference list to be in line with journal formatting requirements.

Reviewer 2 Report

This article examines English- and Spanish-speaking Therapist Behaviors in Parent-Child Interaction Therapy (PCIT).  Authors assert a need for this study due to the fact that PCIT is a best-practice treatment for behavior problems in young children, with prior research illustrating that different therapist coaching styles may be associated with faster skill acquisition and improved parent engagement.  However, the prior research has mostly been conducted among English-speaking families.  Authors contend that there is limited research examining therapist behaviors and treatment delivery among Spanish-speaking clients, therefore establishing a rationale for their study.

The review is as follows:

  1. The introduction provides compelling information to establish a need for the study.
  2. Line 65-97 – There is an insightful, comprehensive discussion of in-vivo training, with good examples provided.
  3. Line 152 – The acronym GANA (Guiando a Niños Activos) should be written out the first time it is introduced.
  4. Line 165- Regarding eligibility for the GANA study, how were participants recruited for the study? What measures were taken to advertise the study to participants?  Were screeners for the study multilingual?  Please expand on this discussion.
  5. Lines 167 and 184-193 - For the participant sample, was there demographic information on the nationalities of the participants?  Authors note that eligibility criteria included if parents identified their child as Mexican American.  Was recruitment particularly for Mexican American families or is it that the sample happened to be composed solely of Mexican American families?  Did participants need to identify as Latinx and/or Mexican American to participate in the study? Where other sociodemographic characteristics collected such as income and occupation?
  6. The findings are interesting and suggest that in Spanish sessions, therapists used significantly unlabeled praises and that Spanish speaking parents gave significantly more direct commands to their children. Is there further insight from the authors as to what contributes to these differential findings to help elucidate results for the reader?

Overall, this is a unique, insightful, and pertinent study.  It is interesting to read.  If authors can attend to clarifying questions about study materials and methodology, this may help improve the paper.

Author Response

We are very appreciative of your careful review of the manuscript, which is now titled, “Examining English- and Spanish-speaking Therapist Behaviors in Parent-Child Interaction Therapy.” The suggestions the reviewers made greatly strengthened the manuscript. We have made the changes in the document you sent with track-changes to ease in review. Responses are outlined below.

  1. The introduction provides compelling information to establish a need for the study. 
  2. Line 65-97 – There is an insightful, comprehensive discussion of in-vivo training, with good examples provided. 

Thank you for these positive reviews.

  1. Line 152 – The acronym GANA (Guiando a Niños Activos) should be written out the first time it is introduced.

The acronym for Guiando a Niños Activos is written out for the first time in line 106 in page 3.

  1. Line 165- Regarding eligibility for the GANA study, how were participants recruited for the study? What measures were taken to advertise the study to participants?  Were screeners for the study multilingual?  Please expand on this discussion. 

We have added the following statements to clarify eligibility for GANA:

Lines 157-161: The research team advertised the study by conducting outreach to programs including Head Start and community mental health and family resource centers to encourage referrals. The study received referrals from various sources: 42% school or teachers, 22% other agency, clinic, or hospital, 12% self-referred, 5% friend, 2% physician, and 2% social worker 

Line 167: Screeners for participants were available in both English and Spanish.

These findings are addressed in the Discussion on page 11, line 451, Finally, all Spanish speaking families in our current study identified as Mexican American. Thus, results found from this sample may not generalize to Spanish-speaking families from other cultural backgrounds.

  1. Lines 167 and 184-193 - For the participant sample, was there demographic information on the nationalities of the participants?  Authors note that eligibility criteria included if parents identified their child as Mexican American.  Was recruitment particularly for Mexican American families or is it that the sample happened to be composed solely of Mexican American families?  Did participants need to identify as Latinx and/or Mexican American to participate in the study? Where other sociodemographic characteristics collected such as income and occupation?

Being Mexican American was an eligibility requirement for the GANA study. We added the following statement on Lines 181-182: “All families from the GANA sample were Mexican American given that this was a criteria for study inclusion.”

  1. The findings are interesting and suggest that in Spanish sessions, therapists used significantly unlabeled praises and that Spanish speaking parents gave significantly more direct commands to their children. Is there further insight from the authors as to what contributes to these differential findings to help elucidate results for the reader?

Thank you for the recommendation to expand on these interesting findings in the Discussion. We have added:

Lines 462-466: Spanish-speaking therapists were found to use higher rates of unlabeled praises. The use of labeled praises as opposed to unlabeled praises may be a point of intervention for training of Spanish-speaking clinicians. Unlabeled praises show warmth, support, and enthusiasm. However, using more specificity, as is the case with clear labeled praises, may better facilitate skill acquisition for parents.

Lines 445-452: Spanish-speaking parents have been found to use higher levels of direct commands in comparison to their English-speaking counterparts, and past literature has offered the possibility of this being due to parents being higher on the control dimension of parenting [38], [39]. This may be helpful in the second phase of treatment, where caregivers are taught to use direct commands with their children. However, Spanish speaking parents may require additional support during the first phase of treatment, given that directive statements, including direct commands, must be avoided during this phase.